# Contemporary Antiretroviral Therapy Dysregulates Iron Transport and Augments Mitochondrial Dysfunction in HIV-Infected Human Microglia and Neural-Lineage Cells

**DOI:** 10.3390/ijms241512242

**Published:** 2023-07-31

**Authors:** Harpreet Kaur, Paige Minchella, David Alvarez-Carbonell, Neeraja Purandare, Vijay K. Nagampalli, Daniel Blankenberg, Todd Hulgan, Mariana Gerschenson, Jonathan Karn, Siddhesh Aras, Asha R. Kallianpur

**Affiliations:** 1Department of Genomic Medicine, Lerner Research Institute, Cleveland Clinic, Cleveland, OH 44195, USA; 2Center for Molecular Medicine and Genetics, Wayne State University, Detroit, MI 48202, USA; 3Department of Microbiology and Molecular Biology, Case Western Reserve University School of Medicine, Cleveland, OH 44106, USA; 4Department of Medicine, Division of Infectious Diseases, Vanderbilt University Medical Center, Nashville, TN 37232, USA; 5Department of Cell and Molecular Biology, John A. Burns School of Medicine, University of Hawaii, Honolulu, HI 96844, USA; 6Department of Molecular Medicine, Cleveland Clinic Lerner College of Medicine of Case Western Reserve University, Cleveland, OH 44195, USA

**Keywords:** HIV, antiretroviral drug, combination antiretroviral therapy, mitochondrial dysfunction, human microglia, iron, neural cell, metabolic reprogramming

## Abstract

HIV-associated cognitive dysfunction during combination antiretroviral therapy (cART) involves mitochondrial dysfunction, but the impact of contemporary cART on chronic metabolic changes in the brain and in latent HIV infection is unclear. We interrogated mitochondrial function in a human microglia (hμglia) cell line harboring inducible HIV provirus and in SH-SY5Y cells after exposure to individual antiretroviral drugs or cART, using the MitoStress assay. cART-induced changes in protein expression, reactive oxygen species (ROS) production, mitochondrial DNA copy number, and cellular iron were also explored. Finally, we evaluated the ability of ROS scavengers or plasmid-mediated overexpression of the antioxidant iron-binding protein, Fth1, to reverse mitochondrial defects. Contemporary antiretroviral drugs, particularly bictegravir, depressed multiple facets of mitochondrial function by 20–30%, with the most pronounced effects in latently infected HIV+ hμglia and SH-SY5Y cells. Latently HIV-infected hμglia exhibited upregulated glycolysis. Increases in total and/or mitochondrial ROS, mitochondrial DNA copy number, and cellular iron accompanied mitochondrial defects in hμglia and SH-SY5Y cells. In SH-SY5Y cells, cART reduced mitochondrial iron–sulfur-cluster-containing supercomplex and subunit expression and increased Nox2 expression. Fth1 overexpression or pre-treatment with N-acetylcysteine prevented cART-induced mitochondrial dysfunction. Contemporary cART impairs mitochondrial bioenergetics in hμglia and SH-SY5Y cells, partly through cellular iron accumulation; some effects differ by HIV latency.

## 1. Introduction

HIV-associated neurocognitive disorder (HAND) remains a common comorbidity of HIV infection, despite combination antiretroviral therapy (cART). cART is highly effective in suppressing HIV replication and restoring immune function, and as a result, life expectancy in people with HIV (PWH) has dramatically increased [1,2]. Among PWH on cART, however, 30–50% still experience cognitive dysfunction, though milder forms of HAND now predominate. HAND adversely impacts quality of life and functional status and carries heightened risks of symptomatic cognitive decline, frailty and death [3,4]. Factors influential in HAND susceptibility include older age, substance use, vascular and other comorbidities, co-infections, lower nadir CD4+ T-cell count, and ancestry, as well as host genetics (e.g., mitochondrial DNA haplogroup) [5,6]. Potential neurotoxicity of cART in the central nervous system (CNS) may also be a factor [7,8,9]. In treated PWH, HAND is characterized by reduced gray and white matter volumes and white-matter microstructural abnormalities on brain imaging studies, and by loss of synaptodendritic complexity histopathologically [10,11].

Within days of infection, HIV-infected blood monocytes breach the blood–brain barrier (BBB), activating perivascular macrophages and microglia, the principal productively infected cells in the CNS [1,12,13]. The establishment of a latent viral reservoir within microglia underlies the persistent neuroinflammation and immune activation characteristic of HAND, which is lessened but not abolished by cART [1,14]. Infected microglia release neurotoxic HIV proteins and inflammatory cytokines and chemokines, which damage astrocytes and neurons, ultimately disrupting neurotransmission [15,16]. CNS neurons may also switch from oxidative metabolism to increased reliance on glycolysis for ATP production, and this “metabolic reprogramming” is due, at least in part, to HIV proteins [17,18]. The establishment of latency in HIV-infected microglia has been associated with decreased glycolytic output, while viral reactivation may reverse this effect [19]. However, the role of metabolic reprogramming in human microglia in HAND pathogenesis during cART is not characterized [18,20].

Concerns that antiretroviral drugs (ARV/s) alone or in combination might adversely affect brain function arose when a randomized clinical trial aimed at reducing the incidence of HAND by treating PWH with highly CNS-penetrating ARVs found poorer neurocognitive performance in recipients of more highly penetrating cART, despite suppressed viral replication in cerebrospinal fluid (CSF) [21]. Another study showing improved cognitive function at 24–96 weeks after cART interruption in PWH with stable HIV infection also suggested the possible neurotoxicity of these drugs [18,22]. The results of subsequent in vitro studies, studies in rodent and non-human primate models, and even recent human post-mortem studies, also support the concept that ARVs can exert adverse metabolic effects in the brain upon long-term exposure [9,18,23]. Unlike older ARVs previously in common usage, contemporary ARVs were thought to be largely free of mitochondrial toxicity, but accumulating data suggest that newer drug classes (e.g., the integrase strand-transfer inhibitors (INSTIs)), could contribute to non-AIDS comorbidities linked to accentuated aging, including HAND, in PWH [9,24]. While HIV Tat and gp120 can disrupt mitochondrial calcium homeostasis and induce mitochondrial dysfunction in glia [18,25], the joint effects of contemporary ARVs and either latent or active HIV infection on mitochondrial function in human CNS-derived cells are not known. This study investigated the ability of contemporary ARVs and cART to induce or augment mitochondrial dysfunction in HIV-infected (HIV+) and HIV-negative, CNS-derived human microglia and in neural-lineage SH-SY5Y cells. Since iron homeostasis is dysregulated in both treated and untreated HIV infection and closely linked to mitochondrial biogenesis and oxidative phosphorylation, we also investigated ARV-related changes in the expression of key cellular and mitochondrial iron-transport proteins in these cell types [5,26,27,28].

## 2. Results

### 2.1. Effects of Individual ARVs or cART on Basal Oxygen Consumption Rate (OCR) in Uninfected (C20) and C20/HIV+ Human Microglia (Hμglia) and in Neural-Lineage SH-SY5Y Cells

Basal OCR values in HIV-uninfected C20 (control) hµglia after 24 h of exposure to lamivudine (3TC), dolutegravir (DTG), and bictegravir (BIC) individually are shown in Figure 1a, as well as a diagram of bioenergetics parameters obtained using the MitoStress assay. In these cells, DTG caused a 76% reduction in basal OCR (median 1.4 vs. 5.8 pmol/min in treated cells vs. vehicle-treated controls, respectively, *p* < 0.0001). As shown in Appendix A, a mixed population of HIV-infected (C20/HIV+), hµglia were sorted into latently infected (GFP-negative) and (re)-activated GFP-positive (GFP+) cell populations. In latently infected HIV+ hµglia (Figure 1b), basal OCR decreased after exposure to 3TC (median 4.3 vs. 5.9 pmol/min in treated cells vs. controls, respectively*, p* < 0.005), DTG (median 3.7 pmol/min, *p* < 0.0001), or BIC (median 4.9 pmol/min, *p* < 0.05). Median basal OCR also decreased in activated HIV+ (GFP+) hµglia following treatment with 3TC (median 5.0 vs. 7.0 pmol/min in treated cells vs. controls, *p* < 0.05), whereas small changes observed after treatment with DTG (median 5.4 pmol/min), and BIC (median 6.2 pmol/min) did not reach statistical significance (Figure 1c). Significant reductions in basal OCR following exposure to emtricitabine (FTC) and tenofovir (TFV), shown in Figure 1d–f, were observed only in uninfected hµglia (both *p* < 0.05).

A reduction of approximately 30% in the basal OCR was observed in SH-SY5Y cells after 24 h of exposure to combined BIC, FTC, and TFV as cART Regimen2 (Figure 1g, mean 29.5 ± 10.4 vs. 41.9 ± 11.9 pmol/min in treated cells and vehicle-treated controls, respectively, *p* < 0.005), which was due to BIC (Figure 1h, mean basal OCR 54.9 ± 3.1 vs. 69.9 ± 6.0 in BIC-treated cells and controls, respectively, *p* < 0.0001). Maximal OCR was 45.4 ± 3.5 (BIC-treated cells) and 61.4 ± 9.5 pmol/min (controls, *p <* 0.0001). The Regimen2 effect persisted at 48 and 72 h of exposure; while basal OCR values after cART Regimen1 (combined DTG, abacavir (ABC), and 3TC) exposure were not significantly lower than in controls at 24 h, these differences also became statistically significant at longer exposures (Appendix A: mean 41.2 ± 8.2 vs. 79.2 ± 9.2 pmol/min in treated and control cells at 48 h, respectively, *p* = 0.007; *p* = 0.0005 at 72 h). No significant changes in basal OCR were observed after 24-h exposures to FTC or TFV individually in SH-SY5Y cells (Figure 1h).

As these initial data showed that Regimen2 causes significant mitochondrial dysfunction in both neural-lineage cells and hµglia at 24 h, all subsequent real-time bioenergetics assays in these cells were performed using Regimen2 ARVs. Although the mitochondrial impact of Regimen2 appeared to be due to BIC alone, all component ARVs were used in further hµglia experiments to account for possible interactions amongst ARVs in the regimen with HIV proteins in cells harboring the provirus; either Regimen2 or BIC alone was used in SH-SY5Y experiments.

### 2.2. Effects of Commonly Used ARVs on Mitochondrial Bioenergetics in Uninfected and HIV+ Hµglia and SH-SY5Y Cells

We next assessed the full spectrum of mitochondrial respiratory function using the MitoStress test following 24-h incubations of uninfected and HIV+ hµglia with combined Regimen2 ARVs. Interestingly, latently infected HIV+ hμglia had the highest basal and overall mitochondrial energy expenditure—significantly higher than activated HIV+ hμglia before treatment—and appeared to be the most affected by contemporary ARVs (Appendix A). In uninfected hµglia, significant reductions were detected at 24 h in the basal OCR, maximal OCR, spare respiratory capacity, ATP production, and proton leak in ARV-exposed vs. untreated cells (all *p*-values < 0.0001, Figure 2).

Results were similar in latently infected HIV+ cells (all *p*-values < 0.0001, Figure 3). In activated HIV+ hµglia, basal and maximal OCR and ATP production were reduced after Regimen2 exposure (all *p-*values *<* 0.001, Figure 4), but not proton leak (Figure 4e); a nonsignificant decline in spare respiratory capacity was also observed (*p =* 0.060). Importantly, in all cell types, non-mitochondrial OCR was also reduced. Mitochondrial OCR values for all three hµglia populations with and without Regimen2 exposure for 24 h are summarized in Appendix A and plotted across groups in Appendix A. As shown in the table, multiple measures of mitochondrial function, including ATP production, were significantly lower in activated HIV+ hµglia compared to latently infected HIV+ hµglia before treatment; exposure to Regimen2 further reduced mitochondrial function in both cell types.

After 48-h incubations of hµglia with Regimen2, all mitochondrial bioenergetics parameters remained lower in uninfected and latently infected HIV+ hµglia (Appendix A, respectively). In activated HIV+ cells, declines in basal OCR and ATP production, while still statistically significant (both *p* < 0.001), were less marked, and maximal respiration compared to vehicle-treated controls was no longer affected (Appendix A). Similar to results at 24 h, at 48 h, proton leak and spare respiratory capacity in activated HIV+ hμglia were not significantly affected by Regimen2 ARVs. In ARV-treated latently infected HIV+ hµglia, basal and maximal OCR and ATP production were also significantly lower at 48 h than at 24 h (all *p* < 0.05; median ATP production 10.2 pmol/min (interquartile range or IQR 4.0–12.6) vs. 13.5 pmol/min (IQR 11.1–16.5), respectively).

Prior reports that neural-lineage SH-SY5Y cells, in fact, have minimal measurable spare respiratory capacity were confirmed in our experiments (Figure 5, *refer also to* Figure 1 *schematic diagram*) [29,30]. Nevertheless, BIC caused a significant decline in basal and maximal OCR at 24 h (Figure 5a,b; both *p* < 0.0001) and a smaller decline in non-mitochondrial OCR compared to vehicle-treated controls. Mitochondrial ATP production rate in the ATP rate assay was also significantly reduced at 24 h after BIC exposure (mean 381.6 ± 93.2 vs. 491.5 ± 63.9 pmol/min in treated vs. control cells, respectively, *p <* 0.005; Figure 5c).

### 2.3. Effects of Contemporary ARVs on ROS Production in Hµglia and SH-SY5Y Cells

Mitochondrial dysfunction is generally associated with increased mitochondrial ROS (mROS) and often total cellular ROS levels [31]. We, therefore, determined Regimen2-induced changes in total cellular ROS and mROS in uninfected and HIV+ hµglia. As shown in Figure 6a,e, we detected no significant changes in total ROS and mROS production after incubation of uninfected hµglia with either Regimen1 or Regimen2 for 24 h. Regimen1 significantly increased total cellular ROS in both latently infected HIV+ and activated HIV+ hµglia (*p =* 0.011 and *p =* 0.012, respectively, Figure 6b,c). Significant increases in mROS, but not total ROS, occurred in latently infected HIV+, but not activated HIV+ or uninfected hµglia treated with Regimen2, and mROS actually declined after Regimen1 exposure in activated HIV+ hµglia (Figure 6e–g). In activated HIV+ hµglia, mROS were also slightly higher with Regimen2 than Regimen1. While total cellular ROS levels were not elevated in hµglia after exposure to Regimen1 or Regimen2 at 48 h (Appendix A), mROS levels were increased in all three hµglia populations after exposure to either cART regimen, particularly in activated HIV+ hµglia (Appendix A). Results for SH-SY5Y cells are shown in Figure 6d,h. Regimen2, as well as BIC alone, increased total cellular ROS levels in these cells (both *p* < 0.01). Mitochondrial ROS levels, as measured by mitoSOX Red, did not differ between Regimen2-treated cells and controls at 24 h but were approximately 40% increased at 36 h (*p* = 1.7 × 10^−8^, Figure 6h).

### 2.4. ARV-Related Changes in Cellular Glycolysis

Glycolytic metabolism, reflected by the extracellular acidification rate (ECAR), was significantly higher in untreated, latently infected HIV+ hµglia, compared to uninfected hµglia (*p* < 0.005, Appendix A). After exposure to Regimen2, the median ECAR in latently infected HIV+ hµglia decreased but remained higher than in treated uninfected hµglia (*p <* 0.05). Notably, the ECAR was not significantly affected by these ARVs in uninfected, latently infected HIV+, or in activated HIV+ cells when compared to ARV-untreated cells of the same type. However, activated HIV+ hµglia treated with Regimen2 had a significantly lower glycolytic rate compared with both treated (*p* < 0.05) and untreated (*p <* 0.005) latently infected HIV+ hµglia. Interestingly, the OCR and ECAR were significantly correlated in all hµglia groups following ARV exposure but not in untreated cells (Appendix A). In SH-SY5Y cells, exposure to Regimen2 or BIC for 24 h did not alter the glycolytic rate compared to controls, as measured by the ATP rate assay (Figure 5c).

### 2.5. Impact of ARVs on Mitochondrial Protein Expression, Morphology, and Mitochondrial DNA (mtDNA) Copy Number

Since ARV exposure for 24 h led to increased cellular and/or mROS production in both hµglia and SH-SY5Y cells, and increased ROS and mitochondrial dysfunction can augment one another [32], we examined the impact of contemporary ARVs on the expression of mitochondrial proteins, including iron–sulfur-containing proteins (Figure 7). Both ARV regimens, particularly Regimen2, significantly suppressed the expression of NDUFS1 (mitochondrial Complex I, Figure 7a–c), SDHB (Complex II, Figure 7d–f), and the Rieske iron–sulfur protein UQCRFS1 (Complex III, Figure 7j–l) in uninfected hµglia. Expression of these proteins was either unaffected or slightly and significantly upregulated in latently infected HIV+ and activated HIV+ hµglia. Expression of Nox2, a key superoxide-generating enzyme involved in cellular ROS production in vivo*,* was not significantly altered by ARVs in hµglia at 24 h (Figure 7g–i). Immunoblot images showing mitochondrial protein expression in the hµglia are shown in Figure 7m. Treatment of SH-SY5Y cells with BIC resulted in decreased NDUFS1 and SDHB expression and increased expression of Translocase of Outer Mitochondrial Membrane 20 (TOM20), a central component of the mitochondrial outer membrane receptor translocase, which is required for translocation of cytosolically synthesized mitochondrial pre-proteins (Figure 7n) [33]. Expression of Nox2 was significantly increased (*p* = 0.020), and mitochondrial superoxide dismutase (Sod2), a superoxide scavenger enzyme, was decreased (*p* = 0.002; Figure 7o). Levels of expression of UQCRFS1 in BIC-treated SH-SY5Y cells were also significantly reduced (Figure 7p), as were levels of mitochondrial supercomplexes I + III2 and III2 + IV (Figure 7q). Defective supercomplex assembly can occur due to disrupted iron–sulfur cluster biogenesis or iron–sulfur protein (e.g., UQCRFS1) formation [34,35]. Finally, levels of the leucine-rich repeat and fibronectin type-III domain-containing protein 2 (Lrfn2), which promotes neurite growth and is a marker of synaptic integrity in the brain [36], were significantly reduced in these neural-origin cells after BIC exposure (Figure 7p).

To determine whether mitochondrial dysfunction due to ARVs translates into defective organellar morphology, confocal microscopy was performed on BIC-exposed SH-SY5Y cells, live-stained with MitoTracker Green. At 24 h following BIC treatment, mitochondria displayed a visibly more fragmented morphology compared to vehicle-treated controls (Figure 8a). MtDNA copy number relative to nuclear DNA was also significantly higher in Regimen2-treated SH-SY5Y cells, compared to controls (*p =* 0.0003, Figure 8b). Compared with uninfected hµglia, latently infected HIV+ and activated HIV+ hµglia both had significantly higher relative mtDNA copy numbers (both *p* < 0.005), but this was reduced by Regimen1 exposure (Figure 8c–f). Treatment with Regimen2 did not appear to alter mtDNA copy number in HIV+ hµglia.

### 2.6. Effects of ARVs on Cellular Iron-Transport Phenotype and Iron Status

The observed changes in the expression of mitochondrial iron-related proteins prompted us to examine ARV-induced changes in the expression of other major proteins involved in cellular iron handling, such as macromolecular ferritin, its ferroxidase-containing subunit (Fth1), TfR, and Fpn. At 24 h, uninfected hμglia did not show significant changes in ferritin expression (Figure 9a). Regimen1 increased ferritin expression (*p <* 0.05) in latently infected HIV+ hμglia, but both regimens significantly reduced ferritin expression in activated HIV+ hμglia (both *p* < 0.05), suggesting a possible interactive effect of ARVs and HIV proteins in the activated HIV+ cells on the expression of ferritin (Figure 9b,c). Non-significant decreases in Fth1 were also detected following Regimen2 treatment (Figure 9d–f,m). TfR and Fpn expression increased after exposure to one or both cART regimens **(**Figure 9g–l), but these differences were statistically significant only in uninfected hμglia (all *p* < 0.05). In SH-SY5Y cells, cell-surface TfR expression was increased at 24 h after BIC exposure, whereas expression of the mitochondrial inner membrane protein mitoferrin, and expression of Fth1, were decreased (Figure 10a–c). In both hμglia and SH-SY5Y cells, however, cellular iron levels were robustly increased by at least 25% and 50%, respectively, after treatment with either Regimen1 or Regimen2, except in activated HIV+ cells, in which iron levels were slightly but significantly decreased (all *p* < 0.05, Figure 10d–g).

Exposure to ARVs for 48 h significantly reduced total ferritin in uninfected hµglia (Regimen1 and Regimen2) and in latently infected HIV+ hµglia (Regimen2 only) but not in activated HIV+ hµglia, when compared to controls (Appendix A). Regimen1 and Regimen2 reduced Fth1 expression in uninfected hµglia, and Regimen1 reduced Fth1 expression in both latently infected HIV+ and activated HIV+ hµglia (Appendix A). TfR expression increased in uninfected hµglia but decreased in latently infected HIV+ hµglia after both regimens, and expression of the iron exporter Fpn decreased and increased, respectively, in latently infected HIV+ and activated HIV+ hµglia exposed to ARVs for 48 h (Appendix A).

Changes in iron homeostasis in all cell types are summarized in Appendix A. Overall, the observed changes show disrupted iron homeostasis in all hµglia cell types, as well as in neural-lineage SH-SY5Y cells, when exposed to ARVs, leading to an inappropriately iron-avid phenotype that would be expected to cause iron-mediated oxidative stress and mitochondrial damage. Since Fth1 levels were consistently downregulated in hμglia and SH-SY5Y cells upon Regimen2 exposure (and prolonged Regimen1 exposure), we next explored whether Fth1 overexpression could rescue the observed ARV-associated mitochondrial defect.

Transfection of hμglia with an Fth-1-expressing plasmid was unsuccessful due to the high efficiency of these cells in degrading exogenous genetic material. However, as shown in Figure 10h, SH-SY5Y cells transfected with an Fth1-expressing plasmid overexpressed Fth1 compared to empty-vehicle controls and no longer showed a reduction in mitochondrial OCR after BIC exposure. These findings suggest that altered iron homeostasis, and specifically, excess labile intracellular iron (which is normally sequestered by ferritins like Fth1), is in part responsible for mitochondrial dysfunction and increased mROS associated with contemporary ARVs. Pre-incubation of SH-SY5Y cells with N-acetylcysteine, a cytosolic ROS scavenger, but not pre-incubation with the effective mROS scavenger, MitoTEMPO, rescued the Regimen2-related mitochondrial defect in these cells (Appendix A), suggesting that cytosolic ROS rather than mROS may be the principal cause of mitochondrial dysfunction in these cells.

## 3. Discussion

Mitochondrial function and energy homeostasis are essential to brain health, given that the brain is responsible for at least 20% of the body’s energy consumption [37]. Steady generation of ATP via mitochondrial oxidative phosphorylation is required to maintain membrane potentials, synaptic transmission, and the integrity of neuronal dendritic architecture [38]. In the absence of an HIV cure, clarifying the metabolic impact of cART in the CNS is of the utmost importance, given the cumulative exposure of PWH to these life-saving medications [39]. Although the microglia HIV reservoir and episodic low-level viral replication (“viral blips”) probably contribute to the high prevalence of cognitive dysfunction seen in virally suppressed PWH, this study highlighted that commonly prescribed ARVs—individually and as components of cART—induce significant metabolic changes in hμglia and neural-lineage (SH-SY5Y) cells, which could also be associated with adverse neurocognitive outcomes. Additionally, the impact of ARVs on mitochondrial function in hμglia varied based on their HIV and HIV latency (activated, GFP+ vs. quiescent transcription, GFP−) status, with generally greater reductions in latently infected HIV+ and uninfected cells than in activated HIV+ cells. Our results also suggest a key role for altered cytosolic and mitochondrial iron homeostasis related to ARVs in the induction of ROS and mitochondrial defects. These observations provide insight into metabolic changes that precede or accompany the neurocognitive decline in virally suppressed PWH on cART and lend credence to the idea that contemporary ARVs not only target mitochondria but may also augment the adverse mitochondrial effects of HIV infection.

Multiple individual ARVs and cART, comprising either DTG, ABC, and 3TC (Regimen1), or BIC, FTC, and TFV (Regimen2), reduced basal OCR in SH-SY5Y cells and/or hμglia, regardless of HIV status. BIC, the primary cause of mitochondrial dysfunction in SH-SY5Y cells treated with Regimen2, and DTG, are both widely prescribed drugs of the INSTI class. At longer drug exposures (i.e., 48 and 72 h), Regimen1 effects were also detectable, and declines in basal OCR more prominent, in SH-SY5Y cells. Major (20–30%) reductions in all facets of mitochondrial respiratory function were detected in both cell types, including maximal respiration, spare respiratory capacity, ATP production, and proton leak. Low proton leak could be due to reduced expression of uncoupling proteins; while increased proton leak has been associated with some diseases, modest mitochondrial uncoupling is important for mitohormesis, the adaptive mitochondrial response to stress that is linked to longevity [40]. Glycolytic rates were not altered by ARVs *per se* in either cell type but instead were linked to the HIV+ status of hμglia: latently infected HIV+ hμglia tended to have the highest ECAR. Therefore, metabolic reprogramming can occur in hμglia harboring provirus. While this finding contrasts with some prior reports that HIV latency is associated with downregulated glycolysis [19], microglia rely on both glycolytic and mitochondrial ATP production, depending on their activation state, and glycolytic metabolism in hμglia has also been linked to their iron status, which we have shown is influenced by ARVs [41,42,43,44]. Findings reported in HIV-infected human primary astrocytes, which, like latently infected HIV+ hμglia sustain little viral replication, were similar to ours, with 20–30% reduction in all mitochondrial parameters [45] compared to uninfected cells and augmented mitochondrial compromise after exposure to the INSTIs DTG or raltegravir; results of ECAR measurements were not reported. The basis of increased basal OCR and other bioenergetic parameters in untreated, latently infected HIV+ hμglia compared to uninfected and/or activated HIV+ hμglia in our study requires further study, but viral suppression during cART has been associated with elevated basal energy expenditure and persistent abnormalities in glycolysis (metabolic programming) [46]. Given that metabolic reprogramming is implicated in both HIV neuropathogenesis and normal aging [18,47], we speculate that the changes we observed may adversely affect the normal metabolic coupling that exists in vivo between microglia and surrounding neurons.

Iron homeostasis is critical for mitochondrial function, and HIV transcription also requires iron [28,35]. Excess iron, however, induces oxidative damage to DNA, lipids, and other macromolecules, as highlighted by the emerging role of ferroptosis (a regulated form of iron-mediated cell death) in diverse diseases [48,49]. HIV infection, as well as certain ARVs, dysregulate iron transport [28,50,51]: the older ARV zidovudine, which remains widely used, alters cellular iron homeostasis and leads to macrocytic anemia in PWH. A recent study also associated other types of cART with anemia, independent of viral suppression, suggesting that other ARV classes can dysregulate iron [27]. Neurons and microglia derive essential iron via receptor-mediated endocytosis of transferrin-bound iron [43], but the antioxidant iron-storage protein Fth1 is required to maintain iron in a non-reactive state and limit the size of the intracellular labile iron pool [52]. We observed dysregulated expression of ferritin, Fth1, TfR and ferroportin in hµglia following ARV exposure and an iron-avid phenotype in SH-SY5Y cells, particularly at 48 h. Both SH-SY5Y cells and hμglia exhibited significantly increased cellular iron content and excess cytosolic and/or mtROS. Such alterations in cellular iron status may also lead to changes in provirus transcription in hμglia and drive metabolic reprogramming during HIV infection [53,54]. In human microglia, increased cellular iron is associated with a more inflammatory phenotype [41,43] and could therefore promote neuroinflammation, despite viral suppression.

In keeping with mitochondrial dysfunction and increased oxidative stress at 24 h of cART exposure and beyond, expression of mitochondrial iron-containing proteins, such as NDUFS1 (Complex I), SDHB (Complex II), and UQCRFS1 (Complex III), was reduced in uninfected hμglia by both ARV regimens and in SH-SY5Y cells by Regimen2 treatment. In SH-SY5Y cells, these changes were accompanied by reduced expression of a synaptic marker associated with cognitive health (Lrfn2), the key mtROS scavenger, Sod2, and mitoferrin (required for iron transport into the mitochondrial matrix for iron–sulfur cluster biogenesis and critical for mitochondrial function). Reduced mitochondrial supercomplex assembly in BIC-treated SH-SY5Y cells also suggests defective iron–sulfur cluster biogenesis in the presence of Regimen2 ARVs, possibly as a result of dysregulated mitochondrial iron transport and altered incorporation of iron into electron-transport-chain complexes. ARV-related downregulation of Fth1 was observed during HIV latency and in SH-SY5Y cells in this study, suggesting a scenario that could promote iron-mediated oxidative stress and mitochondrial injury, as well as glial inflammation [34,35,41]. We also speculate that increased mtDNA copy number, observed in latently infected HIV+ and activated HIV+ hμglia, is a compensatory mechanism for ROS-mediated mtDNA damage and mitochondrial dysfunction. Higher CSF Fth1 levels were recently shown to predict better neurocognitive function over time, particularly in virally suppressed PWH [55]. Our finding that Fth1 overexpression in Regimen2-treated SH-SY5Y cells rescues the basal OCR defect supports the concept that iron-mediated ROS are partly responsible for mitochondrial dysfunction during cART.

Significant lowering of non-mitochondrial OCR also occurred after ARV exposure in all hμglia and neural-lineage cells. This is perplexing and points to a metabolic effect of ARVs that is not confined to mitochondria. Pre-treatment with the cytosolic ROS scavenger N-acetylcysteine but not mitoTEMPO rescued the mitochondrial defect in SH-SY5Y cells, suggesting that cytosolic ROS are upstream of the defect. Regulation of non-mitochondrial respiration, which has both constitutive components and inducible enzymatic components, is understudied, but its reduction could reflect altered extramitochondrial ROS generation [56]. Many cytosolic enzymes, including those of the NADPH oxidase (Nox) family, are involved in cytosolic ROS generation, and Nox2 expression was elevated in BIC/Regimen2-treated SH-SY5Y cells. Nox2 has been implicated in microglia-mediated ROS production and ensuing neuronal injury and death in co-culture studies [57]. Studies in Nox2-knockout mice also demonstrate significantly decreased lipid peroxidation and oxidative stress in the setting of cellular iron accumulation [58], further implicating elevated Nox2 in iron-mediated ROS effects and ferroptosis.

The main limitation of this study is the use of immortalized hμglia and SH-SY5Y cells rather than iPSC-derived human neurons and microglia (or primary cells). The in vitro and in vivo effects of ARVs on these cells may also differ. Nevertheless, hμglia provide a window onto the joint effects of controlled ARV exposure, in the presence or absence of HIV proteins, on cellular metabolism and iron transport. The GFP+ hμglia represent hμglia in which HIV either never became latent after infection, or hμglia in which the virus has reactivated, simulating viral “blips” that are known to occur in PWH on cART. Undifferentiated SH-SY5Y cells also serve as a useful model for the effects of potentially neurotoxic drugs on the developing CNS [59]. Although we did not perform specific tests of cell viability, and this is a limitation, we carefully assessed both SH-SY5Y cells and hμglia just prior to bioenergetic measurements and found no morphologic changes to indicate altered viability. A recent study also reported no impact of several of these ARVs on glial viability in glucose-containing media, even at levels ~100-fold higher than physiologic levels in the CNS [20]. SH-SY5Y cells in our study showed ARV-associated mitochondrial fragmentation, but expression of proteins involved in mitochondrial biogenesis or dynamics, and mitochondrial iron content, were not assessed here. However, sparse information has been available regarding the effects of ARVs other than efavirenz (no longer a first-line ARV) on mitochondrial function in human CNS-derived glia [20,60]. Nor has the metabolic impact of HIV latency or longer ARV exposures in human neural-lineage cells or HIV-infected microglia been adequately studied [61,62,63,64]. The current findings address some of these gaps and provide new clues to the possible role of iron dysregulation in CNS metabolic dysfunction in PWH on cART. Interpretation of these results remains somewhat speculative, however. Further studies should explore substrate utilization, particularly in the absence of glucose, in latently infected HIV+ hμglia in greater detail.

Immunometabolism in the setting of HIV infection is a burgeoning area of research, but few metabolic studies like this one have been undertaken in latently infected or activated HIV+ cells that comprise the HIV reservoir in humans, and we are aware of no such studies in HIV+ human microglia that comprise the CNS reservoir. Prior studies focused on metabolic changes during latency transitions and their impact on infectivity in HIV-infected CD8+ or CD4+ T-cells, not on the metabolic effects of antiretroviral drugs [19,54,65]. Recent gene-expression studies have indirectly evaluated the effects of cART, however: transcriptomic profiles of peripheral blood mononuclear cells (PBMCs) from PWH on suppressive cART may differ significantly from the PBMC profiles in elite HIV controllers, who are naturally virally suppressed without treatment. These transcriptomic studies suggest that inhibition of mitochondrial oxidative phosphorylation contributes to a larger reservoir size during cART [65,66,67]. The changes observed in non-controllers include upregulated glycolysis and glutaminolysis, which are proposed to contribute to accelerated senescence in latently infected immune cells during cART. Most relevant to the current study, Shytaj et al. [54] reported increased expression of antioxidant genes during active HIV replication and upon latency reversal, whereas iron-import pathways were upregulated in association with transition to HIV latency or after cART in PBMCs and CD4+ T-cells from rhesus macaques infected with macrophage-tropic simian immunodeficiency virus. Our study showed similar iron sequestration in CNS-derived hμglia after cART exposure but did not assess the effects of metabolic reprogramming on HIV infectivity, changes in HIV transcription, or deep latency.

In summary, many types of cART impair mitochondrial bioenergetic function in CNS-derived hμglia and SH-SY5Y cells, possibly via altered cellular iron homeostasis in the cytosol and mitochondria and increased iron-mediated ROS. ARVs may also interact with HIV proteins during viral “blips” in latently infected HIV+ microglia and contribute to chronic metabolic changes in the brain. Long-acting ARV formulations [e.g., non-nucleoside reverse-transcriptase inhibitors, INSTIs, and TFV] have been tested in clinical trials, and the metabolic toxicities of these formulations are a concern [68]. Cumulative effects of cART may be particularly impactful in utero or in children and young adults with HIV, who currently require lifelong treatment and whose brain development is ongoing when therapy is initiated [69,70,71,72]. Our results also raise the possibility that chronic cellular iron accumulation due to HIV, and augmented by cART, contribute to cellular senescence and accentuated aging phenotypes in PWH [73,74], including increased brain aging and HAND. Future studies exploring these metabolic changes in greater detail are needed to elucidate underlying mechanisms, which might inform new interventions to prevent and treat cognitive impairment in PWH on cART.

## 4. Materials and Methods

### 4.1. Human Cell Lines and Cell Culture

Human microglia cell lines (hµglia) containing an HIV provirus construct (C20/HIV+) or lacking the construct (uninfected or HIV-negative C20) were kindly provided by Jonathan Karn and used for all in vitro experiments (MTA contract ID# 4127447). The HIV provirus construct in hµglia (replication-incompetent) was tagged with a short-lived green fluorescence protein (GFP) reporter on long terminal repeats, causing cells to fluoresce green upon activation of HIV transcription by pro-inflammatory stimuli [75,76,77,78]. The C20/HIV+ hµglia cell lines containing provirus were sorted into HIV+/GFP-negative (referred to as latently infected HIV+) or HIV+/GFP+ (spontaneously transcribing HIV, referred to as activated HIV+) cells by fluorescence-activated cell sorting (FACS) analysis for all subsequent experiments (Supplemental Appendix A). A Becton-Dickinson LSRFortessa instrument and FlowJo^TM^ V10 software were used for cell sorting and data analysis, respectively.

The hµglia were cultured in Dulbecco’s modified Eagle’s medium (DMEM) with 1% heat-inactivated fetal bovine serum (Gibco—Thermo Fisher Scientific, Waltham, MA, USA), 1X penicillin-streptomycin (Thermo Fisher, Waltham, MA, USA) and 100 µg/mL Normocin^TM^ (InvivoGen, San Diego, CA, USA) at 37 °C in a humidified atmosphere of 5% CO_2_. SH-SY5Y human neuroblastoma cells (lacking HIV provirus) were grown in DMEM-F12 medium with 10% Fetal bovine serum and 1% Penicillin-Streptomycin [59]. The clinical phenotype and viability of hµglia and SH-SY5Y cells were frequently assessed by direct visualization under light microscopy, during ARV incubations and just prior to bioenergetics measurements, and hµglia were passaged no more than 4 times to avoid possible phenotypic evolution.

### 4.2. Cell Culture in the Presence of ARVs

To evaluate the potential effects of different ARVs on real-time mitochondrial function, HIV+ hµglia (FACS-separated into GFP-negative and GFP+ populations), uninfected hµglia, and SH-SY5Y cells were cultured in the presence of one of two contemporary cART regimens for either 24 or 48 h: Regimen1 consisted of DTG (17 ng/mL), ABC (120 ng/mL), 3TC (146 ng/mL), and Regimen2 contained BIC (26 ng/mL), FTC (109 ng/mL), and TFV (50 ng/mL). These commonly administered ARV combinations achieve viral suppression in plasma: they all cross the BBB and similarly suppress HIV replication in CSF [79,80]. Tenofovir was not initially thought to achieve appreciable concentrations in human CSF, but significant TFV levels have been documented in brain tissue after oral dosing (median 148 ng/mL) [81,82]. Concentrations of most of the ARVs tested in hµglia experiments were based on published CSF drug levels reported in PWH who were receiving oral cART [61,83,84,85]. Published data on CSF BIC levels were not available when these experiments were performed, but more recently, median total CSF BIC levels of 10.9 to 11.8 ng/mL have been reported (range 0.5 to 44.9) [86,87]. Incubations of HIV+ and HIV-negative hµglia and SH-SY5Y cells were performed with the same concentrations of individual ARVs/cART.

### 4.3. Real-Time Assays of Mitochondrial Respiratory Function in Intact Cells

For real-time mitochondrial function assays, plating densities for latently infected HIV+ and activated HIV+ hµglia, uninfected hµglia, and SH-SY5Y cells were optimized before performing mitochondrial function measurements. In addition, concentrations of oligomycin (an inhibitor of ATP synthase, or Complex V), carbonyl cyanide 4-trifluoromethoxyphenylhydrazone (FCCP, a potent uncoupler of mitochondrial oxidative phosphorylation) and rotenone/antimycin (inhibitors of Complexes I and III, respectively) were optimized for each cell type and cell line studied, according to the manufacturer’s protocol (MitoStress Assay, Agilent Technologies, Inc., Santa Clara, CA, USA, catalog#103015-100). Concentrations of reagents for in vitro hµglia experiments were as follows: 24 h—oligomycin, 2 μM; FCCP, 2 μM; rotenone/antimycin, 0.5 μM. For 48-h incubations, FCCP was used at 1 μM concentration for activated HIV+ cells. Oligomycin, FCCP, and rotenone/antimycin concentrations used for SH-SY5Y mitochondrial measurements were 1 μM.

We initially investigated the effects of individual ARVs (DTG, ABC, 3TC, BIC, FTC, and TFV, obtained via the NIH Reagent Program, suspended in DMSO) on mitochondrial OCR. Based on these results, as noted below, we performed subsequent experiments using only Regimen2. Mitochondrial function profiles were assessed using a Seahorse XFe96 (SH-SY5Y cells) or XFe24 flux bioanalyzer (hμglia) and commercially available XFe Cell MitoStress Tests Kits (Agilent Technologies, Inc. Santa Clara, CA, USA), according to the vendor’s instructions. Prior to obtaining measurements, on day 1, hµglia were plated at a density of 10^5^ cells/well (and SH-SY5Y cells at 6 × 10^4^ cells/well) in gelatin-coated assay plates in high-glucose DMEM culture medium. On day 2, cells were treated with Regimen2 or with an equal volume of DMSO (vehicle control, or “untreated”); both hµglia and SH-SY5Y cells were incubated with ARVs for 24 or 48 h. Sensor cartridges were hydrated overnight with calibrant solution (MitoStress Kit, Agilent Technologies, Inc., Santa Clara, CA, USA) at 37 °C. On day 3, the culture medium was replaced with an equal volume of fresh Seahorse XF DMEM medium without phenol red at pH 7.4, supplemented with glucose (25 mM for microglia and 10 mM for neurons), 2 mM L-glutamine and 1 mM pyruvate. Cells were maintained at 37 °C in a 5%-CO_2_ incubator for 1 h. The assay medium was again replaced, and the sensor cartridge was loaded with sequential injections of (1) oligomycin, (2) FCCP, and (3) rotenone & antimycin A; mitochondrial OCR readings were obtained. OCR values (in pmol/min) are presented as the mean of the first three measurements, and maximal OCR values were determined from differences in OCR measurements taken after FCCP and rotenone/antimycin A injections. Following the completion of each set of readings, OCR values were normalized to the total cellular protein content within each well. Bioenergetics measurements, including non-mitochondrial oxygen consumption, basal OCR, maximal respiration, proton leak, ATP production, and spare respiratory capacity, were calculated and displayed using Agilent Seahorse^TM^ XF Wave software, version 2.6.1 and Report Generator software, version 4.03; OCR values were expressed in pmol/min. Results are presented as means ± standard deviation (SD) or medians (interquartile range, IQR) from a minimum of six technical replicates and three independent experiments (biological replicates) per condition.

### 4.4. Real-Time ATP Rate Assay (SH-SY5Y Cells)

Mitochondrial and glycolytic ATP production rates were measured on the same Seahorse bioanalyzers. SH-SY5Y cells were plated at a concentration of 6 × 10^4^ cells/well, and assay measurements were performed using the Seahorse XF Real-Time ATP Rate Assay Kit (Agilent, Santa Clara, CA, USA, Catalog #103592-100) and XF DMEM medium, pH 7.4 (Agilent, Catalog #103575-100), with addition of 10 mM glucose, 1 mM pyruvate and 2 mM glutamine as per the manufacturer’s instructions. Briefly, basal OCR and proton efflux rate were measured following serial injections of oligomycin (1.5 μM) and a mix of rotenone and antimycin A (0.5 μM each), which were used for the calculation of the mitochondrial and glycolytic ATP production rates. Bioenergetics values were calculated using the Agilent XF/Seahorse ATP rate assay Report Generator version 4.0.17.

### 4.5. Total and Mitochondrial Reactive Oxygen Species (ROS) Measurements

For mROS assays, hμglia were cultured in 96-well plates (10^5^ cells/well) in high-glucose DMEM. After 24 h, cells were incubated for either 24 or 48 h with ARVs (Regimen1 or Regimen2). SH-SY5Y cells were plated in 24-well or 96-well plates at concentrations of 7.5 × 10^4^ and 1 × 10^4^ cells/well, respectively. The chloromethyl derivative (CM-H_2_DCFDA, Catalog #C6827, Thermo Fisher Scientific) was used as the cellular ROS indicator and MitoSOX Red dye (Catalog #M36008, Fisher) to measure mROS. For total ROS measurement, CM-H_2_DCFDA dye was suspended in 86.5 μL DMSO and added to 13 mLs of DMEM medium (phenol red-free, serum-free, antibiotic-free). After incubation with ARVs, 125 μL of CM-H_2_DCFDA dye was added to each well; two wells were left blank for assessment of background. Plates were wrapped in foil to minimize light exposure. After 60 min of incubation at 37 °C, all medium was aspirated, and fluorescence measurements were taken at excitation filter (*Ex*): 485 nm and emission filter (*Em*): 527 nm. For mROS measurement, the following protocol was followed: after incubation with Regimen1 or Regimen2 for either 24 or 48 h, 125 μL of MitoSOX Red dye (first suspended in 13 μL DMSO and added to 13 μL of phenol-red-free, serum-free, and antibiotic-free DMEM medium) were added to each well, and two wells were retained as background/blank wells. Plates were again incubated in the dark for 20 min at 37 °C; the medium was again removed, and the plates were placed in a microplate reader; *Ex:* 510 nm and *Em:* 580 nm were used for fluorescence measurements. For analyses, readings with dye were subtracted from readings in blank wells (lacking dye). All experiments were performed in triplicate or quadruplicate. Fluorescence measurements were performed for hμglia using a Benchtop FlexStation 3 multi-mode microplate reader and SoftMax^®^ Pro Software version 7.0.3 (Molecular Devices, San Jose, CA, USA) and for SH-SY5Y cells using Synergy H1 microplate reader (Agilent, Santa Clara, CA, USA).

### 4.6. Protein Extraction and Quantification

Total protein was extracted from uninfected, latently infected HIV+, and activated HIV+ hµglia and processed for immunoblots. Cells were washed with 1X phosphate-buffered saline (PBS) and lysed using Radio-Immunoprecipitation Assay (RIPA) buffer. RIPA lysis buffer and EDTA-free Halt^TM^ (Thermo Scientific, Rockford, IL, USA) protease inhibitor cocktail were added to cells for 20 min at 4 °C. Cell lysates were then centrifuged at 10,000× *g* for 30 min at 4 °C, and the supernatant was collected, and protein was quantified by the BCA assay method (Pierce™ BCA Protein Assay Kit, Thermo Fisher). After a 30-min incubation, absorbance was measured at 540 nm in an ELISA plate reader (BIOTEK) using Gen5 software version 3.14.03 (Agilent Inc, Santa Clara, CA, USA). Protein samples were mixed with gel-loading buffer (Laemmli loading buffer containing β-mercaptoethanol as a reducing agent) and boiled for 5 min at 100 °C. Protein samples (50 µg) were separated by SDS-gel electrophoresis using 4–12% Tris-Glycine gradient gels at 4 °C for 2 h at constant 100 V current. Polyvinylidene Fluoride (PVDF) membrane was activated by incubating in methanol for one minute. Separated proteins were then transferred to a pre-activated PVDF membrane at 4 °C for 2 h at a constant 100 V current. For SH-SY5Y cells, protein extractions were also performed using RIPA buffer. Briefly, harvested cell pellets were washed in PBS, lysed in RIPA buffer, and incubated on ice for 15 min. Lysates were then sonicated and centrifuged at 14,000 rpm for 15 min. Supernatants were collected into fresh tubes, and protein quantitation was performed using the Quick Start Bradford protein assay (Biorad Cat #5000201), as previously described [88,89].

Membranes incubated overnight at 4 °C in freshly prepared blocking buffer (5% blotting grade blocker, 500 mM Tris, 1.5 M NaCl, 0.5% Tween 20, pH 7.4). After blocking, membranes were probed with primary antibodies against iron-transport proteins of interest: anti-ferritin (rabbit polyclonal antibody, 1:1000, F5012, Sigma Aldrich, St. Louis, MO, USA), anti-ferritin heavy-chain 1 (Fth1, rabbit monoclonal antibody, 1:1000, 4393S, Cell Signaling Technology, Danvers, MA, USA), anti-ferroportin 1 (Fpn, rabbit polyclonal, 1:500, NBP1-21502, Novus Biologicals, Centennial, CO, USA), and anti-β-actin (mouse monoclonal, 1:5000; MAB1501, MilliporeSigma, Burlington, MA, USA). Antibodies against other proteins, including mitochondrial proteins of interest, included: anti-leucine rich repeat and fibronectin type III domain-containing 2 (Lrfn2, rabbit polyclonal antibody, 1:500, ab106366, Abcam, Waltham, MA, USA) and anti-ubiquinol-cytochrome c reductase Rieske iron-sulfur polypeptide 1 (UQCRFS1, rabbit polyclonal antibody, 1:500, 18443-1-AP, Proteintech, Chicago, IL, USA). Other antibodies used were: NADPH oxidase 2 (Nox2 rabbit polyclonal antibody, 1:500, 19013-1-AP, Proteintech, Chicago, IL, USA), mitochondrial superoxide dismutase 2, (Sod2, rabbit polyclonal antibody, 1:500, 24127-1, Proteintech, USA), NADH: ubiquinone oxidoreductase core subunit S1 (NDUFS1, rabbit polyclonal antibody, 1:500, 12444-1-AP, Proteintech, USA), succinate dehydrogenase complex iron-sulfur subunit B (SDHB, rabbit polyclonal antibody, 1:500, 10620-1-AP, Proteintech, USA), mitoferrin (rabbit polyclonal antibody, 1:500, 26469-1-AP Proteintech, USA), Tom20 antibody (rabbit monoclonal antibody, 1:500, 42,406, Cell Signaling Technology, Danvers, MA, USA), CD71/transferrin receptor (TfR, rabbit monoclonal antibody, 1:500, 13113, Cell Signaling Technology, USA), and tubulin (rabbit monoclonal antibody, 1:500, 9099, Cell Signaling Technology, USA) overnight at 4 °C. Membranes were washed and then incubated with horseradish peroxidase (HRP)-conjugated secondary anti-rabbit antibody (for iron-related proteins, 1:40,000, NA931V, GE Healthcare, Buckinghamshire, UK) and anti-rabbit antibody (1:40,000, NA934V, GE Healthcare, UK) for 1 h at room temperature, and for mitochondrial proteins, with 1:5000 antibody, at room temperature for 2 h). Membranes were incubated for 5 min in chemiluminescence substrate reagents (luminol and peroxidase) (Thermo Scientific SuperSignal West Pico PLUS Chemiluminescent, Waltham, MA, USA). Bands were visualized, and band intensities were measured and analyzed using the Invitrogen^TM^ iBright^TM^ imaging system and software (ThermoFisher Scientific, Waltham, MA, USA).

### 4.7. Mitochondrial DNA (MtDNA) Copy Number Estimation and Morphologic Assessment

Total genomic DNA was isolated from cells using the Invitrogen PureLink Genomic DNA Mini Kit (Thermo Fisher Scientific, K1820-01). Real-time PCR analysis was performed using the SYBR Green assay on an ABI 7500 PCR system (Applied Biosystems). Data were analyzed using the ΔΔCt method. The primer sequences used to amplify mtDNA (*MT-TL1* gene) and GAPDH were as follows: mtDNA: forward, 5′-CCTCCCTGTACGAAAGGAC-3′; reverse, 5′- GCGATTAGAATGGGTACAATG-3′; GAPDH: forward, 5′-GAGTCAACGGATTTGGTCGT-3′; reverse, 5′-TTGATTTTGGAGGGATCTCG-3′.

Mitochondrial morphology was assessed using MitoTracker Green FM (Thermo Fisher Scientific, M7154) as per the manufacturer’s instructions. Briefly, cells were plated on coverslips and incubated with 200 nM dye for 30 min. Cells were rinsed once with 1X PBS and imaged at Ex*:* 490 nm/Em 516 nm. Cells were imaged on a Leica TCS SP5 microscope, and images were processed using Adobe Photoshop, version 24.0.

### 4.8. Transient Transfection of SH-SY5Y Cells and Other Pre-ARV-Treatment Interventions

SH-SY5Y cells were transfected with the indicated plasmids using TransFast transfection reagent (Promega, Madison, WI, USA, E2431) according to the manufacturer’s protocol. The empty vector (EV) plasmid used was the backbone vector that did not harbor the target gene. The amount of plasmid was determined based on the size of the culture plate, as per the TransFast technical manual; a TransFast: DNA ratio of 2:1 in a complete medium was used. After incubation at room temperature for ~15 min, cells were overlaid with the mixture. Following overnight incubation, the medium was completely replaced, and the plates were incubated further for the indicated time.

The ability of N-acetylcysteine (NAC) and MitoTEMPO to reduce or reverse ARV-induced mitochondrial defects in SH-SY5Y cells was also tested by a brief (10-min) pre-incubation of cells with these ROS scavengers (NAC, for total cellular ROS and Mito-TEMPO for mROS), at concentrations of 100 μM and 20 μM, respectively, along with Regimen2.

### 4.9. Statistical Analyses

Data were analyzed using STATA statistical software version 17.0 (StataCorp LLC, College Station, TX, USA) and represented as dot plots, showing the medians and interquartile ranges (IQRs) of mitochondrial OCR measurements, using GraphPad Prism, version 9.5.1 (GraphPad Software, Inc., La Jolla, CA, USA). Results for untreated control and treated cells were compared, and *p*-values were calculated using the (non-parametric) Wilcoxon Rank Sum (Mann–Whitney U) test. Comparisons of protein bands by densitometry were performed similarly. The strength of associations between OCR and ECAR values in hμglia was assessed using non-parametric Spearman’s Rank correlations.

## Figures and Tables

**Figure 1 ijms-24-12242-f001:**
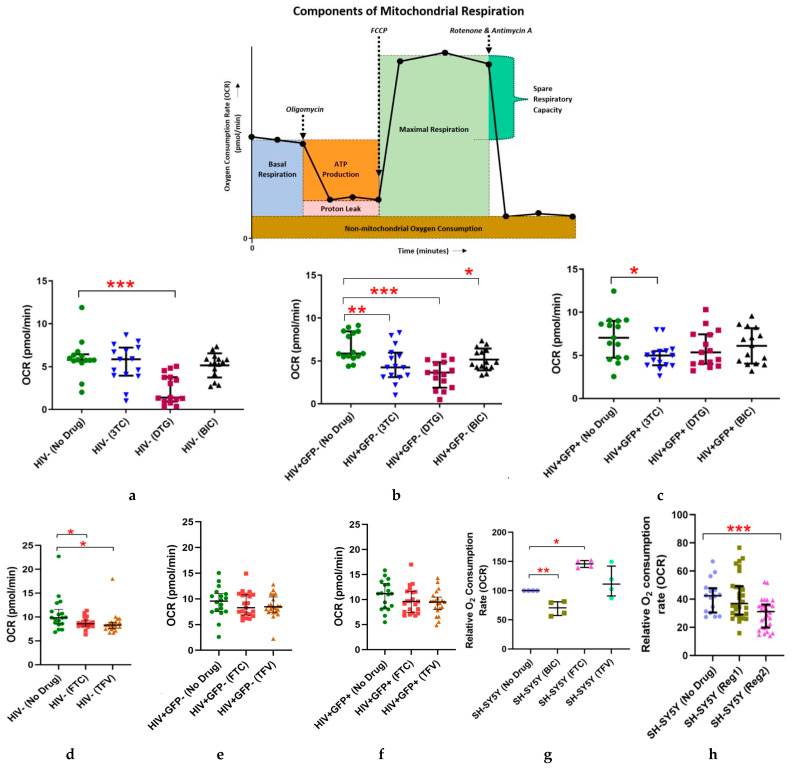
Effects of a 24-h incubation with individual antiretroviral drugs on the basal oxygen consumption rate (*OCR*) in: (**a**,**d**) uninfected hμglia, (**b**,**e**) latently infected HIV+ hμglia, and (**c**,**f**) activated HIV+ hμglia. (**g**,**h**), Effects of a 24-h incubation with drugs individually (**g**) or in combination (**h**) in SH-SY5Y neural-lineage cells. Median and interquartile ranges of OCR values and results of at least 3 independent experiments (each with a minimum of 6 technical replicates) are shown. *Abbreviations: Reg* (Regimen)1, dolutegravir, abacavir, lamivudine; *Reg2*, bictegravir, emtricitabine, tenofovir; *3TC*, lamivudine; *DTG*, dolutegravir; *BIC*, bictegravir; *TFV*, tenofovir; *FTC*, emtricitabine. * *p* < 0.05; ** *p* < 0.005; *** *p <* 0.0001 (Mann–Whitney U test).

**Figure 2 ijms-24-12242-f002:**
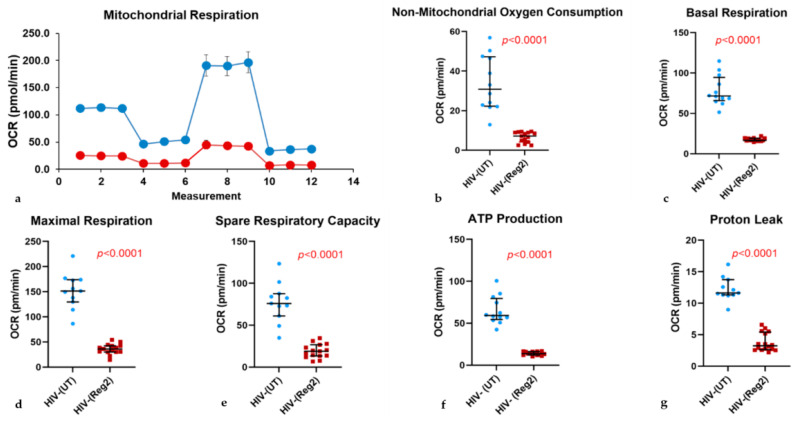
Effects of a 24-h incubation of HIV-uninfected hμglia with Regimen2 (*Reg2*), showing (**a**) the MitoStress curve (blue = vehicle-treated controls; red = drug-treated cells), (**b**) non-mitochondrial oxygen consumption rate (*OCR*), (**c**) basal respiration, (**d**) maximal respiration, (**e**) spare respiratory capacity; (**f**) ATP production, and (**g**) proton leak. Median OCR values and interquartile ranges in treated vs. control (*UT*) cells from at least six technical replicates and three separate experiments are shown. (*p*-values obtained by Mann–Whitney U test).

**Figure 3 ijms-24-12242-f003:**
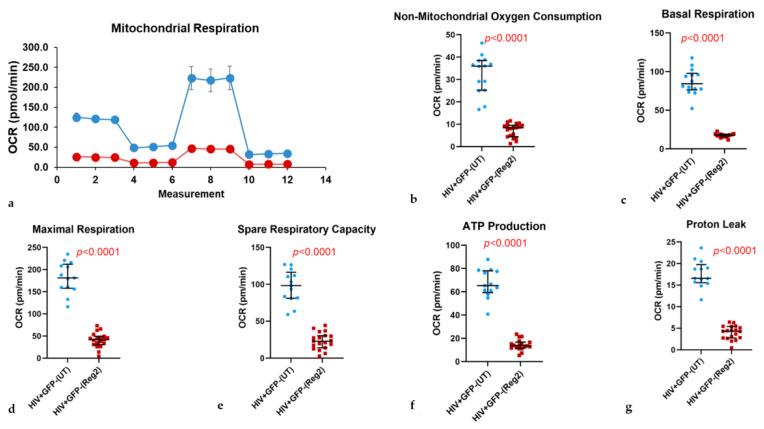
A 24-h incubation of latently infected HIV+ hμglia with antiretroviral Regimen2 (*Reg2*) significantly reduces all mitochondrial respiratory parameters, including non-mitochondrial oxygen consumption rate (*OCR*). (**a**) MitoStress curve (blue = vehicle-treated controls; red = drug-treated cells); (**b**) non-mitochondrial OCR; (**c**) basal respiration; (**d**) maximal respiration; (**e**) spare respiratory capacity; (**f**) ATP production; (**g**) proton leak. Median OCR values and interquartile ranges in treated vs. control (*UT*) cells from at least six technical replicates and three independent experiments are shown. (*p*-values obtained by Mann–Whitney U test*)*.

**Figure 4 ijms-24-12242-f004:**
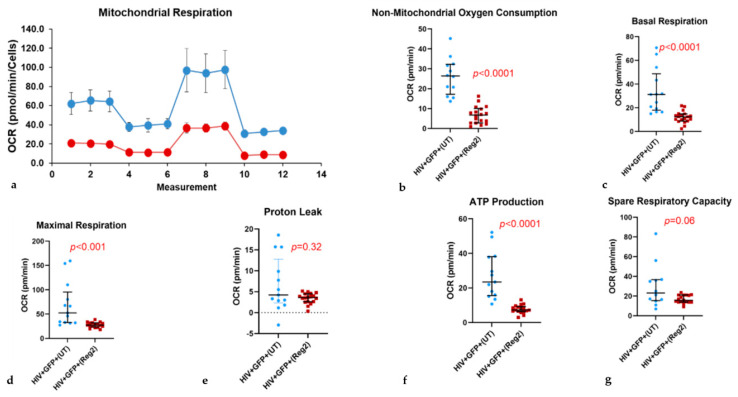
In activated HIV+ hμglia, the antiretroviral drug Regimen2 (*Reg2*) reduces multiple components of mitochondrial respiratory function at 24 h. (**a**) MitoStress curve (blue = vehicle-treated controls; red = drug-treated cells); (**b**) non-mitochondrial OCR; (**c**) basal respiration; (**d**) maximal respiration; (**e**) proton leak; (**f**) ATP production; (**g**) spare respiratory capacity. Median OCR values and interquartile ranges in treated vs. control (*UT*) cells from at least six technical replicates and three independent experiments are shown. (*p*-values obtained by Mann–Whitney U test).

**Figure 5 ijms-24-12242-f005:**
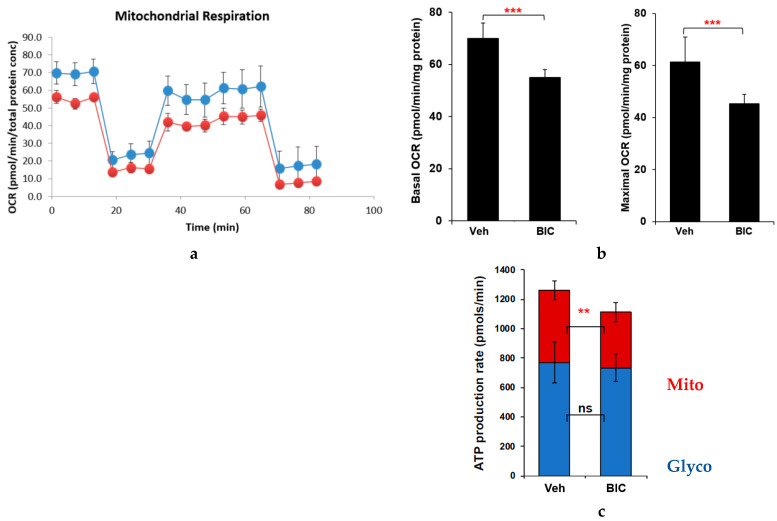
Effects of a 24-incubation with BIC on (**a**) mitochondrial respiratory function in the MitoStress test (blue = vehicle-treated controls; red = drug-treated cells), and (**b**) basal and maximal OCR, and (**c**) ATP production, measured in the ATP rate assay, in SH-SY5Y cells. *Abbreviations: OCR*, oxygen consumption rate; *Veh*, vehicle-treated controls; *BIC*, bictegravir-treated cells; *Mito*, mitochondrial ATP production; *Glyco*, glycolytic ATP production; *ns*, not statistically significant. ** *p* < 0.005; *** *p* < 0.0001.

**Figure 6 ijms-24-12242-f006:**
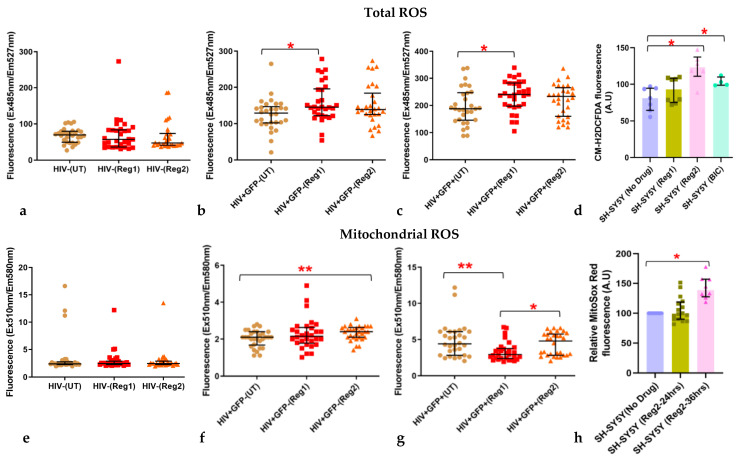
Effects of Regimen1 and Regimen2 ARVs after 24 h on (**a**–**d**) total cellular and (**e**–**h**) mitochondrial ROS in (**a**,**e**) uninfected, (**b**,**f**) latently infected HIV+, and (**c**,**g**) activated HIV+ hμglia and in (**d**,**h**) SH-SY5Y cells. *Abbreviations: UT*, untreated/vehicle controls; *Reg1*, Regimen1; *Reg2*, Regimen2; *BIC*, Bictegravir. * *p <* 0.05; ** *p <* 0.005 (Mann–Whitney U test).

**Figure 7 ijms-24-12242-f007:**
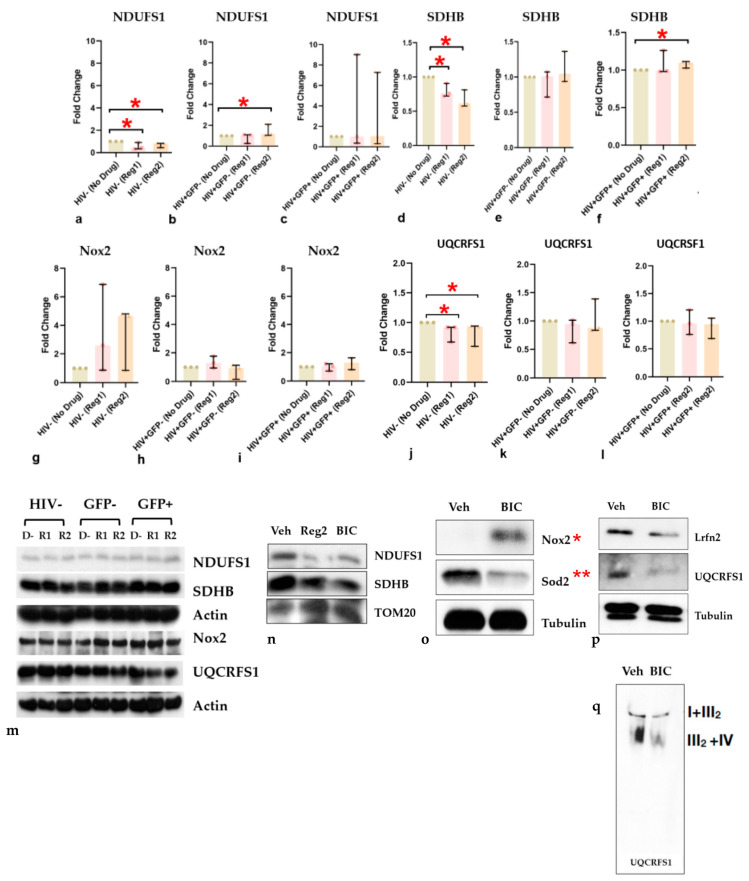
(**a**–**l**) Expression of NDUFS1, SDHB, Nox2, and UQCRFS1 in uninfected, latently infected HIV+(GFP−), and activated HIV+(GFP+) hμglia after exposure to Regimens1 and 2 for 24 h. (**m**) Immunoblot images of protein expression in hμglia and (**n**–**q**) SH-SY5Y cells. *Abbreviations: D−*, no drug (vehicle control); *R1*, Regimen1; *R2,* Regimen2. NDUFS1, SDHB, and UQCRFS1 are mitochondrial respiratory subunit proteins; *Sod2*, a mitochondrial ROS-scavenging enzyme; *Nox2*, NADPH oxidase 2, a major cytoplasmic oxidase; *Tom20*, a mitochondrial protein translocase; *Lrfn2*, a marker of neuronal/synaptic integrity; *I + III_2_* and *III_2_ + I*V, mitochondrial supercomplexes. * *p <* 0.05; ** *p <* 0.005 (Mann–Whitney U test).

**Figure 8 ijms-24-12242-f008:**
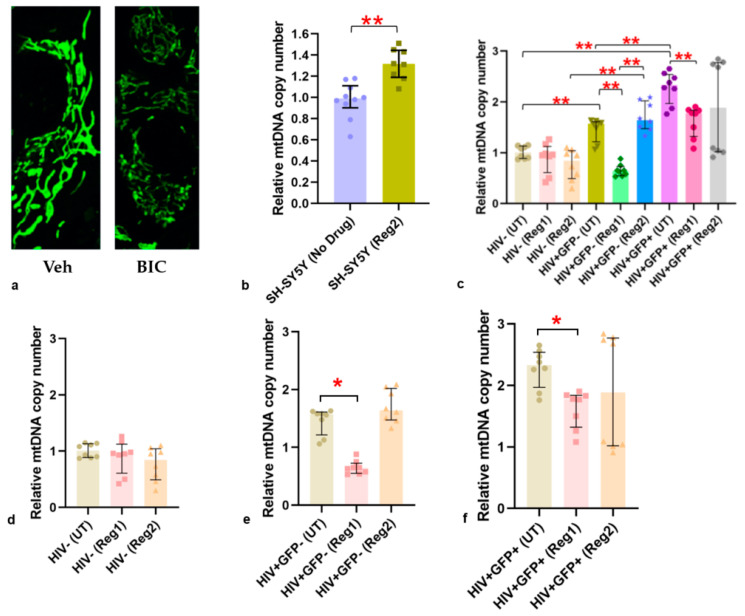
(**a**) Mitotracker green image of mitochondrial fragmentation in SH-SY5Y cells after 24 h of bictegravir (*BIC*) exposure, compared with vehicle-treated controls (*Veh*). (**b**) Relative mitochondrial DNA copy number in Regimen2-treated vs. vehicle (no drug) in SH-SY5Y cells. (**c**–**f**) Relative mtDNA copy number in hμglia: (**d**) HIV-uninfected (HIV−), (**e**) latently infected HIV+(GFP−), and (**f**) activated HIV+(GFP+) hμglia. *Abbreviations: UT*, vehicle-treated controls; *Reg1*, Regimen1; *Reg2*, Regimen2; *mtDNA*, mitochondrial DNA. * *p <* 0.05; ** *p <* 0.005 (Mann–Whitney U test).

**Figure 9 ijms-24-12242-f009:**
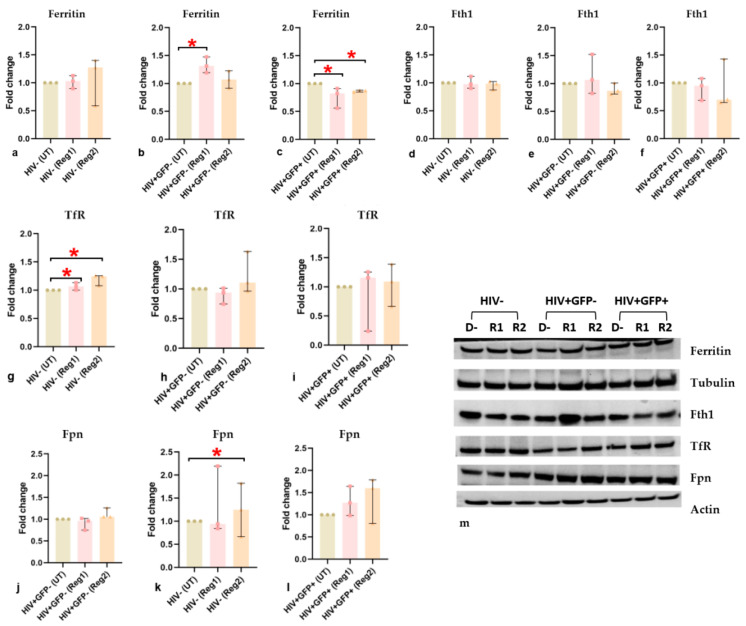
(**a**–**c**) Quantification of total ferritin, (**d**–**f**) ferritin heavy-chain (*Fth1*), (**g**–**i**) transferrin receptor (*TfR*), and (**j**–**l**) ferroportin (*Fpn*) in uninfected, latently infected HIV+(GFP−), and activated HIV+(GFP+) hμglia, after 24-h exposure to vehicle (*UT* or *D−*) or ARVs in Regimen1 (*Reg1*, *R1*) or Regimen2 (*Reg2*, *R2*). (**m**) Immunoblot gel images showing iron-related protein expression in hμglia. Results of a minimum of 3 separate experiments are shown. * *p <* 0.05 (Mann–Whitney U test).

**Figure 10 ijms-24-12242-f010:**
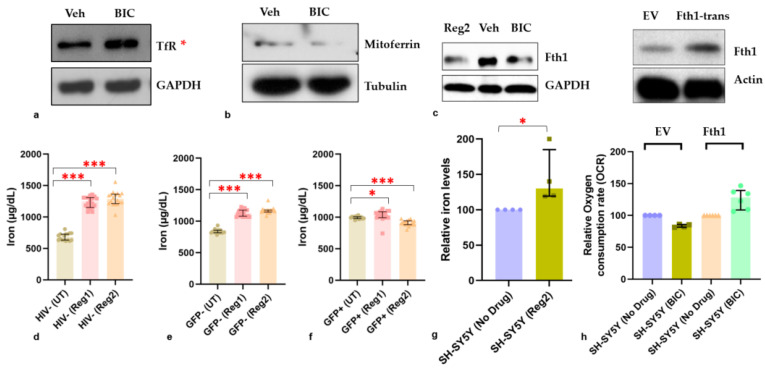
Iron dysregulation in hμglia and SH-SY5Y cells after 24 h of exposure to Regimen1 (*Reg1*) or Regimen2 (*Reg2*) ARVs, or bictegravir (*BIC*). (**a**) Changes in TfR, (**b**) Mitoferrin, and (**c**) Fth1 expression in SH-SY5Y cells after BIC. Changes in cellular iron in (**d**) uninfected, (**e**) latently infected HIV+(GFP−), and (**f**) activated HIV+(GFP+) hμglia and (**g**,**h**) SH-SY5Y cells. *Fth1*, ferritin heavy-chain 1, *Fth1-trans*, cells transfected with Fth1-expressing plasmid; *TfR*, transferrin receptor; *EV*, empty plasmid vector; *Veh*, Vehicle. * *p <* 0.05; *** *p* < 0.005 (Mann–Whitney U test).

## Data Availability

Data are contained within the manuscript. Experimental raw data supporting the reported results will be made available upon request.

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
