# Peer review of "Contemporary Antiretroviral Therapy Dysregulates Iron Transport and Augments Mitochondrial Dysfunction in HIV-Infected Human Microglia and Neural-Lineage Cells"

_ijms, 2023, doi:10.3390/ijms241512242_

Round 1

Reviewer 1 Report

This is a very interesting study and well presented. I would accept the paper in its current form. 

Reviewer 2 Report

The article by Kaur and colleagues is well written and structured.

In section 4.6 of the materials and methods, the references Aras 2015 and Aras 2013 are not mentioned correctly.

Reviewer 3 Report

Comments for the Authors

ijms-2442460

Contemporary Antiretroviral Therapy Dysregulates Iron Transport and Augments Mitochondrial Dysfunction in HIV-Infected Human Microglia and Neural-Lineage Cells

Harpreet Kaur, Paige Minchella, David Alvarez-Carbonell, Neeraja Purandare, Vijay K. Nagampalli, Daniel Blankenberg, Todd Hulgan, Mariana Gerschenson, Jonathan Karn, Siddhesh Aras, Asha R. Kallianpur

The main theme of this manuscript is to study the impact of different combinations of contemporary antiretroviral drugs (Regimen 1 and Regimen 2) on mitochondrial dysfunction using human microglia (hµglia) cell line harboring inducible HIV proviruses and SH-SY5Y cells. Kaur and colleagues measured and compared parameters, including reactive oxygen species (ROS) production (Figures 1-6), glycolysis (Supplemental Figure 7), mitochondrial DNA copy number (Figure 8) representing the metabolic activity and expression of mitochondrial proteins (Figure 7) and the proteins involved in cellular iron metabolism and regulation (Figures 9 and 10) in uninfected hµglia cells, hµglia cells with latent and active infections and in SH-SY5Y cells. Kaur and colleagues claimed that impairment of mitochondrial bioenergetic functions in hµglia cells and SH-SY5Y cells can result from the changes of cellular iron homeostasis, mitochondrial dysfunction, and increased production of ROS after contemporary antiretroviral drugs are given, and the cause of this damage is more relevant to the effects of antiretroviral drugs themselves than HIV infections.

This manuscript consists of lots of data but lacks concise data representation and discussion/conclusion, rendering this manuscript somehow hard to follow. Major and minor comments are listed as follows:

Major comments:

1. Immunometabolism and HIV-1 pathogenesis represent a new area of HIV-1 research. Several studies relevant to this topic have been remarkable progress (Valle-Casuso et al, 2019, Cell Metabolism; Shytaj et al, 2020, the EMBO Journal; Shytaj et al, 2021, EMBO Molecular Medicine…etc) using CD4+ T cells. It is known that HIV-1 reservoir in macrophages differs from CD4+ T cells in many aspects, especially the authors investigating a specific type of macrophage (microglia) present in the brain. Thus before investigating the influence of antiretroviral drugs on the metabolic activity of HIV-1-infected microglia cells, it could be valuable if the authors may first characterize whether different levels of metabolic activity (ROS and ECAR measurements for example) in microglia cells affect HIV-1 pathogenesis (i.g. infectivity, transcription..etc) in general. If it has been done, please summarize the current status of the studies concerning the metabolic activity/reprogramming in HIV-1 reservoir cells in the brain.

2. The authors tested the impact of antiretroviral drugs on the neuroblastoma cells, SH-SY5Y in the absence of viral infection or HIV-1 proteins. In this case, the purpose to include this cell line here is not clear to me because SH-SY5Y should not be infected by HIV-1 in nature. And how can results obtained from experiments using SH-SY5Y be linked to those using hµglia in this study and HIV-1 infections in the brain in general?

3. Hµglia with latent infections (GFP-negative cells) isolated from cells with active infections (GFP-positive cells) was based on spontaneous activation (Supplemental Figure 1). Did the authors verify whether the proportion between GFP-negative and GFP-positive hµglia remains similar in every round of spontaneous activation or whether the proportion varies? If the stochastic phenotype is observed, it most likely means that additional factors in hµglia affect HIV gene expression. In this case, can the authors comment on whether such fluctuations in HIV transcription (latent vs active infections) may affect the impact of antiviral drugs (Regmen 1 and 2) on mitochondrial dysfunction? 

In addition, after antiretroviral drugs were given, did GFP-negative cells remain transcriptionally inactive and so did GFP-positive cells?

4. On pages 4 and 5, the authors stated that latently infected-HIV+ hµglia had higher basal and overall mitochondrial energy expenditure than others. Please cite the Figure corresponding to this statement. At least I cannot see an obvious difference comparing Figure 2c and 3c.

5. In line with the previous comment, on page 5, the second paragraph, the authors mentioned that proton leak was not affected under Regimen 2 in activated HIV+ hµglia. Please cite the corresponding Figure. So does the sentence “ATP production, were significantly lower in … compared to latently infected-HIV+ hµglia at 24 hours” in the same paragraph. Please cite the corresponding Figure.

6. On page 6, what was the minimal measurable spare respiratory capacity referred to in this case? And where to see this confirmation in Figure 5?

7. Mitochondrial ATP production rate is a critical parameter to measure and I cannot find that the ATP rate assay was done using hµglia harboring viral infections.

8. Please carefully discuss the different phenotypes of total cellular and mitochondrial ROS (Figure 6) caused by drugs Regimen 1 and Regimen 2 between cells with active and latent infections. Whether different combinations of drugs and the transcriptional status of HIV affect the cellular metabolism (ROS) in this case.

9. In supplemental Figure 7, how do the authors interpret that latently infected hµglia showed a high basal level of ECAR measurement (untreated)? And please discuss the biological meaning that the ECAR measurement only showed a decrease in cells with active infections.

10. Relevant to the previous comment, can the authors plot the correlation between OCR measurement and ECAR measurement among uninfected- and latently infected hµglia and hµglia harboring active infections? In other words, can cells harboring latent and active infections be well separated based on cellular or mitochondrial metabolic activity?

11. The authors claimed that both ARV regimens significantly suppressed the expression of NDUFS1, SDHB, and UQCRFS1 in HIV-infected and/or uninfected hµglia (section 2.4.). However, NDUFS1 in HIV+GFP+, SDHB in HIV+GFP-, and UQCRFS1 in both HIV+GFP+ and HIV+GFP- showed no statistical difference compared with cells without infections. In addition, it is ambiguous as the authors wrote nonsignificantly increased concerning expression of Nox2. Overall, the protein expression patterns are not consistent for HIV+GFP- and HIV+GFP+ in the presence of Regmen 1 and 2. It seems to be hard for me to bring a solid conclusion based on these results.

12. The conclusion made by the authors concerning Figure 9a-9c (section 2.5.) somehow escapes me. What HIV proteins are referred to here? It requires a more detailed explanation.

13. Can the authors comment on the finding shown in Supplemental Figure 8 that if disrupted iron homeostasis might be associated with the transcriptional status of proviruses?

14. On page 14, line 4, “ARVs on mitochondrial function in hµgliavaried based on their HIV latency status,”, what do the authors mean by HIV latency status? Are we talking about the transcriptional status here or HIV latency and deep latency?

Minor comments:

1. It is recommended to plot grouped dot/box plots for Figure 2 to Figure 4. It is quite different to compare the mitochondrial metabolic activity among uninfected-, and latently infected hµglia and hµglia with active infections in the current version of the manuscript.

Round 2

Reviewer 3 Report

Comments for the Authors

ijms-2442460

Contemporary Antiretroviral Therapy Dysregulates Iron Transport and Augments Mitochondrial Dysfunction in HIV-Infected Human Microglia and Neural-Lineage Cells

Harpreet Kaur, Paige Minchella, David Alvarez-Carbonell, Neeraja Purandare, Vijay K. Nagampalli, Daniel Blankenberg, Todd Hulgan, Mariana Gerschenson, Jonathan Karn, Siddhesh Aras, Asha R. Kallianpur

I appreciate the authors’ efforts to answer my questions point-by-point. Their detailed responses were fair and very clear to me. The quality and clarity of this revised version of the manuscript have been greatly improved. I do not have further comments/questions concerning this current version of the manuscript.